# Domain adaptation for supervised integration of scRNA-seq data

Yutong Sun[1] & Peng Qiu [2✉]

Large-scale scRNA-seq studies typically generate data in batches, which often induce non-trivial batch effects that need to be corrected. Given the global efforts for building cell atlases and the increasing number of annotated scRNA-seq datasets accumulated, we propose a supervised strategy for scRNA-seq data integration called SIDA (**S**upervised **I**ntegration using **D**omain **A**daptation), which uses the cell type annotations to guide the integration of diverse batches. The supervised strategy is based on domain adaptation that was initially proposed in the computer vision field. We demonstrate that SIDA is able to generate comprehensive reference datasets that lead to improved accuracy in automated cell type mapping analyses.

[1] School of Electrical and Computer Engineering, Georgia Institute of Technology, Atlanta, Georgia, USA. [2] Department of Biomedical Engineering, Georgia Institute of Technology and Emory University, Atlanta, Georgia, USA. ✉email: peng.qiu@bme.gatech.edu

Given the increasing use of scRNA-seq and the global efforts for building cell atlases in various biological contexts, large amounts of scRNA-seq data are generated and accumulated. Large-scale scRNA-seq studies typically generate data in batches, where samples are processed at different time points, handled by different personnel and labs, prepared and sequenced by different technology platforms, which are all potential causes of batch effects[1]. It is well-documented that batch effect is often strong in scRNA-seq, making it challenging to effectively integrate multiple scRNA-seq batches around a common biological theme (e.g., tissues, organs) into a single comprehensive atlas that fully captures the heterogeneity of the biological theme[2]. In the literature, a majority of the existing scRNA-seq batch integration methods address batch effects in an unsupervised manner, aligning the distributions of cells across different batches. A few examples of popular methods include canonical correlation analysis (SeuratV2[3]), mutual nearest neighbor approach (MNN Correct[4] and fastMNN[5], Scanorama[6], BBKNN[7]), nonnegative matrix factorization (LIGER[8]), and variational autoencoder (scVI[9], scGen[10]). These unsupervised methods assume that many cell types are shared among the datasets to be integrated and run into the risk of aligning distinct cell types when the assumption does not hold.

Since many scRNA-seq datasets come with clustering analysis and cell type annotations performed by the researchers who generated the data, there is an opportunity to perform supervised data integration, using the cell type annotations to inform the data integration. By construction, supervised integration should outperform unsupervised approaches because the cell type annotations can be used to encourage cells with the same annotations across different batches to overlap and encourage cells with different annotations to be separated. A few supervised integration algorithms have been proposed recently. scAlign[11] is an integration algorithm that provides a fully supervised option called scAlign+, which trains a deep neural encoder that incorporates cell type labels to map functionally similar cells to the same coordinates in a latent representation space. LAmbDA[12] trains a classifier that maps cell type labels and removes batch effects by constructing a label mask that determines the known relationships between the cell type labels of two batches. SMNN[13] and iSMNN[14] perform batch effect correction via supervised mutual nearest neighbor detection. In this study, we are interested in developing a supervised integration algorithm that can compete with existing state-of-art integration algorithms for scRNA-seq data. In computer science, one way to implement supervised integration is supervised domain adaptation. In general, domain adaptation is to leverage information to a target domain from a different but related source domain, where the domains can be different batches in the context of scRNA-seq data integration.

The discussion of unsupervised and supervised approaches for scRNA-seq data integration is distinct from that in the context of automated cell type mapping, which is a related computational problem. Automated cell type mapping is typically supervised, where machine learning models are constructed from either prior knowledge of cell type marker genes[15] or previously annotated scRNA-seq reference datasets[16–19] and then applied to annotate cells in newly generated query datasets. The machine learning models could be based on invariant similarity metrics such as correlations[16], tree-based classifiers[15,17], or nearest neighbor classifiers[18,19], etc. The nearest neighbor approach for cell type mapping is sometimes referred to as cell type label transfer and is typically implemented by unsupervised integration of reference and query data without considering cell type annotations in the reference data, followed by supervised classification that uses cell type annotations of the reference data to label cells in the query data. Therefore, although a majority of existing cell type mapping

algorithms are supervised, supervised approaches for scRNA-seq data integration are less explored in the existing literature.

In this paper, we developed a supervised scRNA-seq data integration algorithm using a domain adaptation deep neural network called SIDA (**S**upervised **I**ntegration using **D**omain **A**daptation). Given multiple scRNA-seq batches to be integrated, we implemented the Siamese Network[20] to learn a shared embedding space that integrates multiple batches. The learning objective is a combination of contrastive semantic alignment loss and classification loss. We compared SIDA with three unsupervised scRNA-seq data integration algorithms in a recent benchmark study[1], including SeuratV3[21] and Harmony[22], which ranked highest in the benchmarking study, as well as limma[23], which ranked relatively lowly in the benchmarking study. In addition to the unsupervised algorithms, we also compared SIDA with two supervised integration algorithms, scAlign+[11] and LAmbDA[12]. Since scAlign+ also provides an unsupervised option (scAlign), we included both the supervised and unsupervised implementations of scAlign for completeness. According to the evaluation metric in ref. [24], SIDA provided significantly improved performance over both the existing unsupervised and supervised algorithms. Intuitively, the improved performance of supervised integration over unsupervised integration was expected because the supervised approach used additional information on cell type labels to inform the integration. However, among the three supervised integration algorithms, SIDA achieved overall remarkable improvement compared to the unsupervised integration algorithms. The improvement of SIDA over the best unsupervised algorithm was larger than the range of performance among the unsupervised algorithms, suggesting that scRNA-seq data integration should be performed in a supervised fashion whenever possible. To further demonstrate the utility of SIDA, we evaluated the integrated data in terms of its ability to serve as reference data for automated cell type mapping algorithms. We showed that SIDA generated more comprehensive references that led to improved cell type mapping accuracy for new datasets.

## Results

**SIDA framework**. To achieve supervised integration, we propose to use a domain adaptation deep learning network architecture, which is able to incorporate cell type labels to inform data integration. As shown in Fig. 1, this network architecture takes training pairs generated by cells from different batches as input and passes the input cells through two identical network branches, "g" with shared weights, projecting the cells into a common embedding space. The network and weights are trained to optimize the classification and contrastive semantic alignment loss, which includes a semantic alignment loss that minimizes the distance between cells from different batches but of the same cell type, a separation loss that maximizes the distance between cells from different domains and cell types, and a classification loss encourages high classification accuracy, and hence further maximizing the distance between cells of different cell types, which further facilitate the integration process and the clustering of different cell types. Given a scRNA-seq data collection of multiple batches along with cell type labels of the cells, we sample cell pairs from the batches to train the proposed domain adaptation deep learning network, which is able to produce an embedding space where the batch effect is minimized based on both the distribution of the data and the cell type labels. Details of the design are described in the "Methods" section.

**Data collection for evaluation**. We evaluate SIDA on five collections of scRNA-seq data in the contexts of the pancreas, PBMC, gut, pancreatic islet, and hematopoietic stem cells

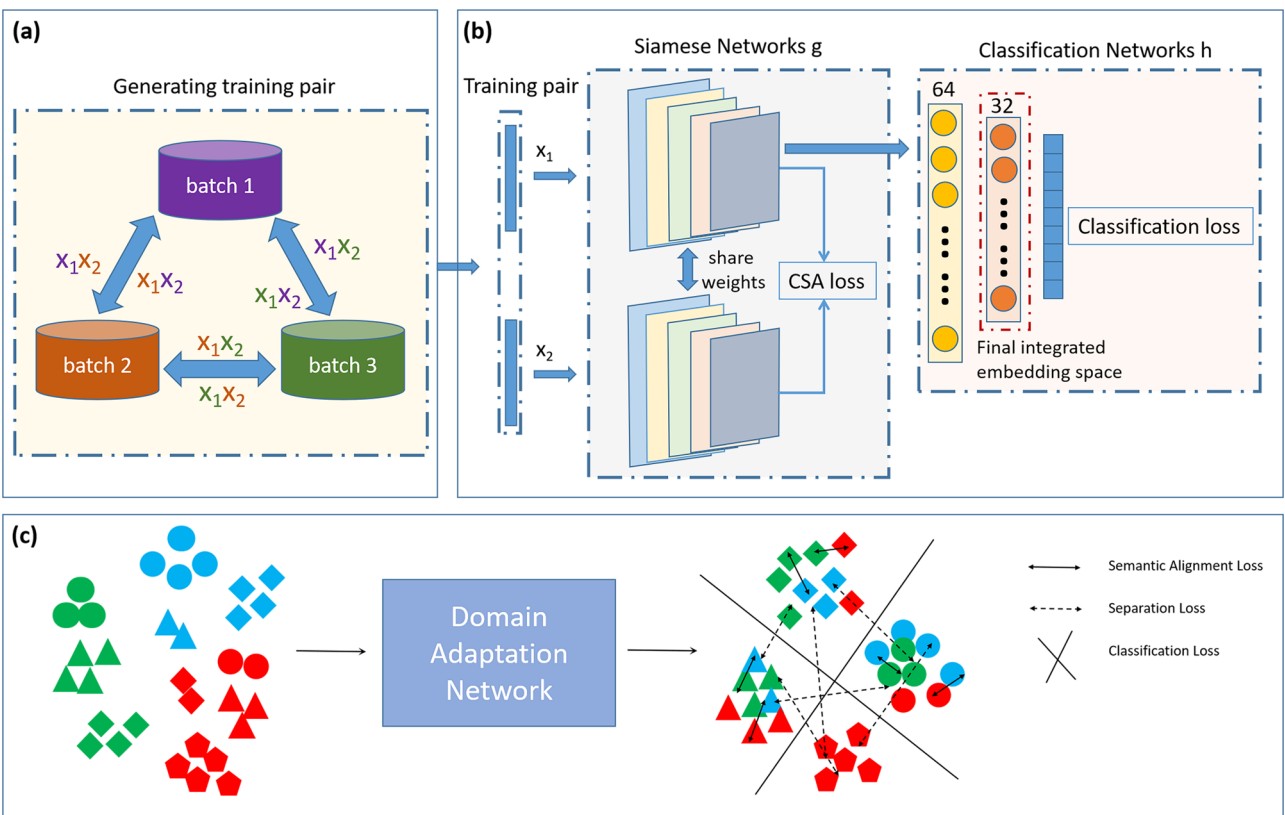

**Fig. 1 Overview of the SIDA algorithm. a** Generating cell pairs for training network model. **b** Domain adaptation network structure. **c** Classification and contrastive semantic alignment loss. Training pairs of cells from different batches are fed into a convolution network "g" and projected to a shared embedding space to optimize semantic alignment loss and separation loss. The embedded data are fed into a classification network "h" to optimize the classification loss.

(HSCs). Each collection contains multiple datasets, which we consider batches. The pancreas data collection consists of five batches of human pancreatic cells, including Baron[25], Mutaro[26], Segerstolpe[27], Wang[28], and Xin[1,29]. In total, there are 15 different cell types across the batches, among which four cell types appear in all batches, and four cell types appear in only one of the batches. The PBMC data collection consists of five batches of peripheral blood mononuclear cells, including control, stim[30], PBMC3k[31], 10x 3′, and 10x 5′[1,32]. These batches contain a total of 11 different cell types, with 5 appearing in all batches and 1 being batch specific. The gut data collection consists of four batches, Bigaeva[33], Huang[34], Parikh[35], and Wang[36]. These four batches contain 11 different cell types in total, among which 3 cell types appear in all batches, and 1 cell type is batch specific. The pancreatic islet data collection consists of four batches of human pancreatic islet cells, including CEL-Seq, CEL-Seq2, Fluidigm C1, and Smart-Seq2[11,26]. These four batches contain 13 different cell types in total. Since all these 13 cell types appear across all four batches, this pancreatic islet data collection does not contain any batch-specific cell type. The HSCs data collection consists of two batches of hematopoietic stem cells[11]. These two batches contain three different cell types in total, which appear across both batches, meaning that the HSCs data collection does not contain batch-specific cell type. Detailed references to these data collections and individual datasets are provided in Supplementary Note 1 and Supplementary Table 1.

**SIDA leads to improved batch mixing and cell type separation.** We applied SIDA, four unsupervised integration methods (SeuratV3[21], Harmony[22], limma[23], scAlign[11]) and two supervised

integration methods (scAlign+[11], LAmbDA[12]) to three data collections (pancreas, PBMC, gut), generating integrated versions for each data collection separately. The integrated datasets are evaluated in terms of both batch mixing and cell type separation. We use a k-nearest neighbor-based approach to define positive rate and true positive rate, which quantify batch mixing and cell type separation[24]. We also examined evaluation metrics used in a recent benchmark paper for scRNA-seq data integration[1], including k-nearest neighbor batch-effect test (kBET), local inverse Simpson's index (LISI), average silhouette width (ASW), and adjusted rand index (ARI). Details of these evaluation metrics are described in the "Methods" section.

For the pancreas data collection, the integration results are shown in the tSNE visualizations in Fig. 2a, b, colored by cell types and batch labels. Seurat and Harmony successfully mixed the different batches, as shown in the second and third columns in Fig. 2b. However, when colored by cell type labels, the second and third columns of Fig. 2a show that Seurat and Harmony improperly aligned some of the distinct cell types in different batches, e.g., stellate and mesenchymal, acinar and ductal. From the fourth, fifth, and sixth columns of Fig. 2a, b, we can observe that Limma, scAlign, and scAlign+ performed poorly, where the same cell type in different batches did not align and mix with each other. LAmbDA successfully aggregated the same cell type and mixed the different batches. However, the last column in Fig. 2b shows that LAmbDA did not separate different cell types properly. As shown in the first column of Fig. 2a, b, SIDA was able to correctly align corresponding cell types across batches and separate different cell types. As a quantitative comparison of the three supervised and the four unsupervised algorithms on the pancreas data collection, Fig. 3a shows six evaluation metrics for each algorithm. Among the

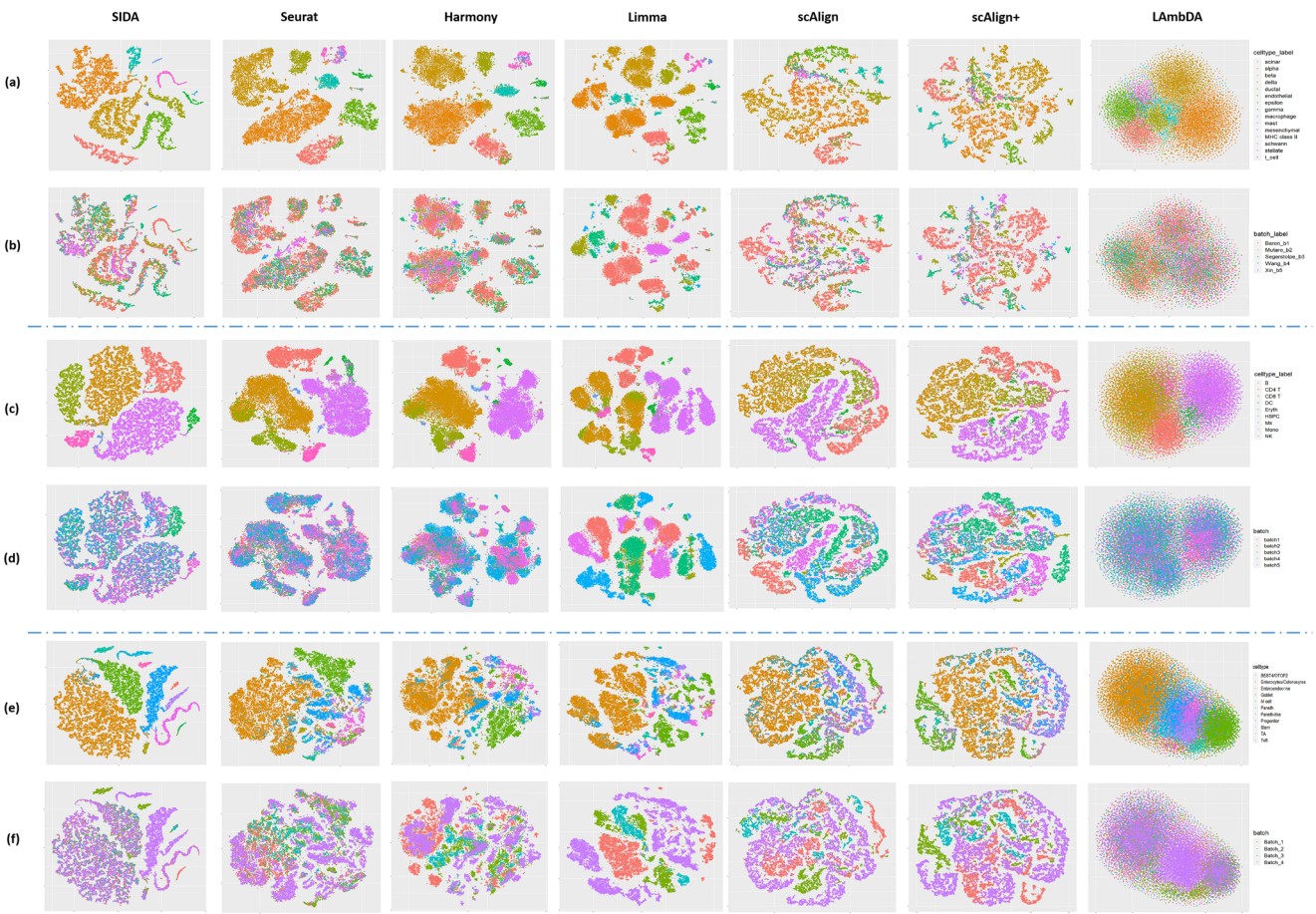

**Fig. 2 tSNE visualization of SIDA, four unsupervised algorithms (Seurat, Harmony, Limma, scAlign) and two supervised algorithms (scAlign+ and LAmbDA) applied to three data collections. a, b** Integration of pancreas data collection colored by cell types and batch labels; **c, d** Integration of PBMC data collection colored by cell types and batch labels; **e, f** Integration of gut data collection colored by cell types and batch labels.

six metrics, SIDA achieved the highest performance for four metrics, the second highest for kBET, and the third highest for LISI. This quantitative evaluation shows that SIDA achieved better cell type separation and batch mixing compared to the four unsupervised and the two supervised methods, which is consistent with the visualization results in Fig. 2a, b.

For the PBMC data collection, the integration results are shown in the tSNE visualization in Fig. 2c, d, colored by cell types and batch labels. The first column of Fig. 2c, d shows that SIDA performed well on this more difficult PBMC data collection, achieving proper mixing of different batches. Based on the second and third columns of Fig. 2c, d, Seurat and Harmony mixed the different batches, but Seurat and Harmony improperly aligned two similar cell types: CD4 T and CD8 T. Based on the fourth, fifth, and sixth columns of Fig. 2c, d, we observe that Limma, scAlign, and scAlign+ failed to properly integrate the PBMC data collection, which is consistent with their performance in the pancreas data collection. From the last column of Fig. 2c, d, we can observe that LAmbDA did not separate different cell types properly. As a quantitative comparison of the supervised and unsupervised integration algorithms in the PBMC data collection, Fig. 3b shows the six evaluation metrics for each algorithm in the PBMC data collection. SIDA achieved the highest performances for four metrics and the second highest for kBET and LISI, among which improvement in the true positive rate was the most significant.

Integration results for the gut data collection are shown in the tSNE visualization in Fig. 2e, f, colored by cell types and batch

labels. As shown in the second to sixth columns of Fig. 2f, Seurat, Harmony, Limma, scAlign, and scAlign+ did not effectively mix the batches and, therefore, did not properly align corresponding cell types in different batches, as shown in Fig. 2e. LAmbDA successfully mixed the four different batches as shown in the last column of Fig. 2f. However, when colored by cell type labels, the last column of Fig. 2e shows that LambDA improperly aligned different cell types. In contrast, tSNE visualization of SIDA showed desirable batch mixing, alignment of corresponding cell types in different batches, as well as separation among different cell types. The performance difference shown in the tSNE visualizations was also reflected in the quantitative comparison shown in Fig. 3c, where SIDA consistently achieved the highest performance for five metrics and the second highest for kBET. Moreover, according to all six metrics, except for the positive rate, the improvement of SIDA over the best unsupervised algorithm was larger than the range of performance among the four unsupervised algorithms.

In addition to the tSNE plots in Fig. 2, we also visualized the integration results using UMAP shown in Supplementary Note 3 and Supplementary Fig. 1, where the observations and interpretations are highly consistent with the tSNE visualizations. Since both tSNE and UMAP are nonlinear dimension reduction tools to visualize high-dimensional distributions in two-dimensional space, the numerical values and range axes of tSNE and UMAP plots are not interpretable. Therefore, we removed the axis labeling of tSNE and UMAP plots, following the practice in a previous benchmarking paper for scRNA-seq data

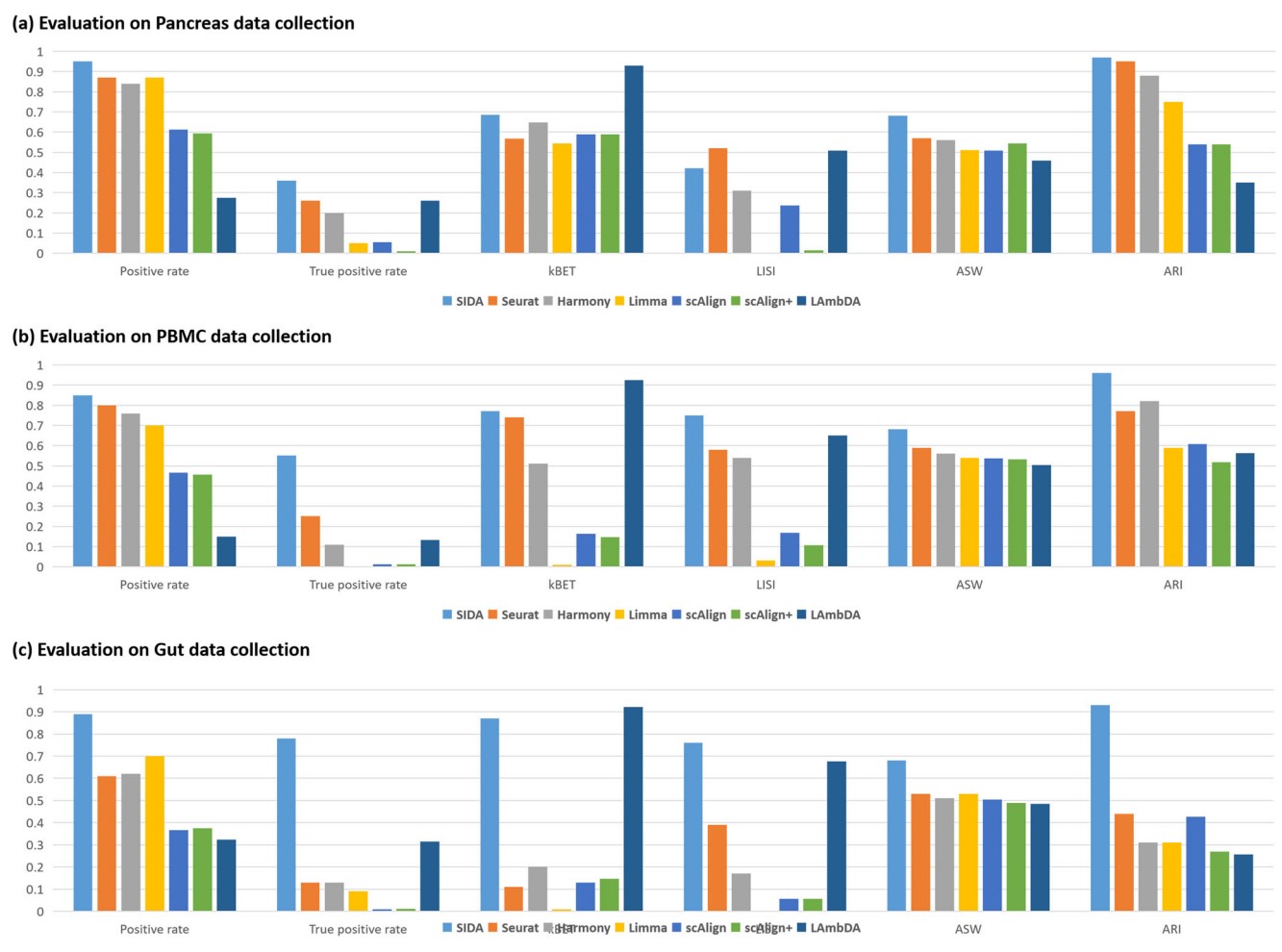

**Fig. 3 Comparing supervised and unsupervised integration algorithms using six quantitative evaluation metrics for batch mixing and cell type separation. a** Evaluation based on the pancreas data collections; **b** Evaluation based on the PBMC data collection; **c** Evaluation based on the gut data collection.

integration[1]. The numerical values of the quantitative metrics in Fig. 3 are summarized in Table 1. Comparing the evaluation metrics across the three data collections, we observed that all integration algorithms performed well on the pancreas data collection, whereas the integration performance was slightly lower in the PBMC data collection and the lowest in the gut data collection. Therefore, it seemed that the pancreas, PBMC, and gut data collections were progressively more and more challenging to integrate. It was encouraging to observe that SIDA achieved more pronounced performance improvement over existing unsupervised and supervised algorithms in the PBMC and gut data collections that were relatively more challenging to integrate.

**Comparison between SIDA and supervised scAlign+.** To further demonstrate the strength of SIDA, we performed an additional comparison with scAlign, which provides both unsupervised (scAlign) and supervised (scAlign+) options[11]. We performed the comparison on the pancreas islet and the HSCs data collections, which were used in scAlign's tutorial demonstrations (https://github.com/quon-titative-biology/scAlign). We examined these data collections to make sure that we were able to faithfully reproduce the integration results in scAlign's tutorial demonstrations, which would ensure a fair comparison with SIDA. For completeness, our comparison included both unsupervised and supervised options of scAlign. Figure 4a, b shows the tSNE visualizations of integration results of the HSCs data

collection, colored by cell types and batch labels. Since the HSCs data collection consists of only two batches and all three cell types in the data collection appeared in both batches, it presented a relatively simple data integration challenge. Based on the tSNE visualizations in Fig. 4a, b, all three algorithms achieved decent integration performance on this HSCs data collection, aligning shared cell types across the two batches, among which SIDA and scAlign+ more significantly separated different cell types. UMAP visualizations shown in Supplementary Fig. 2 provided the same observation and interpretation. In Fig. 4c, a comparison based on the six quantitative metrics showed that SIDA achieved the best performance in most metrics except for LISI, which shows the effectiveness of SIDA over scAlign and scAlign+. The second and third columns of Fig. 4a–c show that supervised scAlign+ achieved significantly improved performance compared to the unsupervised scAlign, which is consistent with the intuition that supervised integration is able to improve batch mixing and cell type separation in scRNA-seq data integration.

Figure 5a, b shows the tSNE visualizations of integration results of the pancreatic islet data collection, colored by cell types and batch labels. Figure 5b shows that all three algorithms were able to generate embedding spaces where cells in various batches were mixed together. In the first panel of Fig. 5a colored by cell types, SIDA successfully delineated various cell types in the data. However, the remaining two panels of Fig. 5a show that scAlign and scAlign+ were not as effective in properly separating distinct cell types. For example, the ductal cell type was split into two

**Table 1 Comparing supervised and unsupervised integration algorithms using six quantitative evaluation metrics for batch mixing and cell type separation.**

|          |          | Positive rate | True positive rate | kBET     | LISI     | ASW      | ARI      |
| -------- | -------- | ------------- | ------------------ | -------- | -------- | -------- | -------- |
| Pancreas | SIDA     | **0.95**      | **0.36**           | 0.69     | 0.42     | **0.68** | **0.97** |
|          | Seurat   | 0.87          | 0.26               | 0.57     | **0.52** | 0.57     | 0.95     |
|          | Harmony  | 0.84          | 0.2                | 0.65     | 0.31     | 0.56     | 0.88     |
|          | Limma    | 0.87          | 0.05               | 0.54     | 0        | 0.51     | 0.75     |
|          | scAlign  | 0.61          | 0.06               | 0.59     | 0.24     | 0.51     | 0.54     |
|          | scAlign+ | 0.59          | 0.01               | 0.59     | 0.01     | 0.54     | 0.54     |
|          | LAmbDA   | 0.28          | 0.26               | **0.93** | 0.51     | 0.46     | 0.35     |
| PBMC     | SIDA     | **0.85**      | **0.55**           | 0.77     | **0.75** | **0.68** | **0.96** |
|          | Seurat   | 0.8           | 0.25               | 0.74     | 0.58     | 0.59     | 0.77     |
|          | Harmony  | 0.76          | 0.11               | 0.51     | 0.54     | 0.56     | 0.82     |
|          | Limma    | 0.7           | 0                  | 0.01     | 0.03     | 0.54     | 0.59     |
|          | scAlign  | 0.47          | 0.003              | 0.16     | 0.17     | 0.54     | 0.61     |
|          | scAlign+ | 0.46          | 0.003              | 0.15     | 0.11     | 0.53     | 0.52     |
|          | LAmbDA   | 0.15          | 0.13               | **0.93** | 0.65     | 0.50     | 0.56     |
| Gut      | SIDA     | **0.89**      | **0.78**           | 0.87     | **0.76** | **0.68** | **0.93** |
|          | Seurat   | 0.61          | 0.13               | 0.11     | 0.39     | 0.53     | 0.44     |
|          | Harmony  | 0.62          | 0.13               | 0.2      | 0.17     | 0.51     | 0.31     |
|          | Limma    | 0.7           | 0.09               | 0.01     | 0        | 0.53     | 0.31     |
|          | scAlign  | 0.37          | 0.008              | 0.13     | 0.06     | 0.51     | 0.43     |
|          | scAlign+ | 0.38          | 0.01               | 0.15     | 0.06     | 0.49     | 0.27     |
|          | LAmbDA   | 0.32          | 0.32               | **0.92** | 0.68     | 0.49     | 0.26     |

Bold indicates the best performance for each metric in each data collection.

islands far away from each other in the embedding space. For alpha, beta, and delta cell types, cells were co-located in close proximity but separated in multiple small islands, where the distance between islands corresponding to different cell types could be smaller than the distance between islands corresponding to the same cell type. UMAP visualizations of this comparison shown in Supplementary Fig. 3 provided the same observation and interpretation. The integration performance in terms of batch mixing and cell type separation is also reflected in the quantitative comparison shown in Fig. 5c. Interestingly, Fig. 5c shows that supervised scAlign+ achieved minimal improvement over unsupervised scAlign when integrating this pancreatic islet data collection, which was a relatively more difficult integration challenge that involved multiple batches with a nontrivial number of cell types. Meanwhile, SIDA consistently achieved significant improvements over scAlign and scAlign+ across all six quantitative evaluation metrics, which indicates the effectiveness and robustness of SIDA.

**SIDA improves the accuracy of automated cell type mapping**. To demonstrate the utility of SIDA in terms of cell type annotation, we applied a leave-one-out strategy to each data collection. For a given data collection, we first left out one batch and integrated the remaining batches using either SIDA or an existing integration algorithm. We then performed automated cell type mapping to predict the cell type labels of the left-out batch using the integrated data as a reference. The resulting cell type mapping accuracy was used to evaluate which integration algorithm was able to build a more comprehensive reference that led to better performance in cell type annotation of left-out data that was not used to generate the integrated data.

For an integrated dataset to serve as the reference in automated cell type mapping, the integrated data in the low-dimensional embedding space was insufficient. Instead, we needed to convert the integrated data from the low-dimensional embedding space back to the original high-dimensional gene space. To achieve this, we picked one of the batches in the integrated data as the target space, applied the Mutual Nearest Neighbors strategy in Seurat to

find anchors between the picked batch and the other batches in the low-dimensional embedding space, and used weighted differences of the anchors in the original gene space to convert the integrated low-dimensional data to the original high-dimensional gene space, so that the integrated data in high-dimensional gene space resembled the picked batch. When converting the integrated low-dimensional space to the high-dimensional gene space, we could pick any of the integrated batches as the target space; therefore, one integration algorithm produced several integrated versions of integrated data, and the number of versions was the same as the number of batches that were integrated. After generating an integrated dataset using one integration algorithm with one choice for the target space, the integrated dataset was considered as the reference data for cell type mapping, and the left-out batch was considered as the query data. We applied the cell mapping pipeline in scanpy[37], which first selected high variable genes and then used a PCA-based function to predict the cell type labels for the query cells based on the reference data. Figure 6 shows the results of the cell type mapping.

Cell type mapping in the pancreas data collection is relatively simple. The first row of the heatmaps in Fig. 6a shows cell type mapping accuracies between each pair of individual batches in the pancreas data collection. When Wang or Xin served as the reference, the accuracies were lower compared to cases where the other three batches served as the reference. This variation in performance was expected because certain batches may not be sufficiently comprehensive to serve as the reference for cell type mapping. In the remaining rows of the heatmaps in Fig. 6a, we generated integrated data by different algorithms and converted the data to different choices of target space. Based on the average of each row across all heatmaps in Fig. 6a, we observed that all three supervised integration algorithms showed similar performance, around 6–8% improvement in cell mapping accuracy compared to individual batches serving as reference. Among the unsupervised integration methods, scAlign achieved a 6% improvement, whereas the other three only achieved a 1–2% improvement. Such improvement was more noticeable for target spaces defined by Wang and Xin. Therefore, starting with a

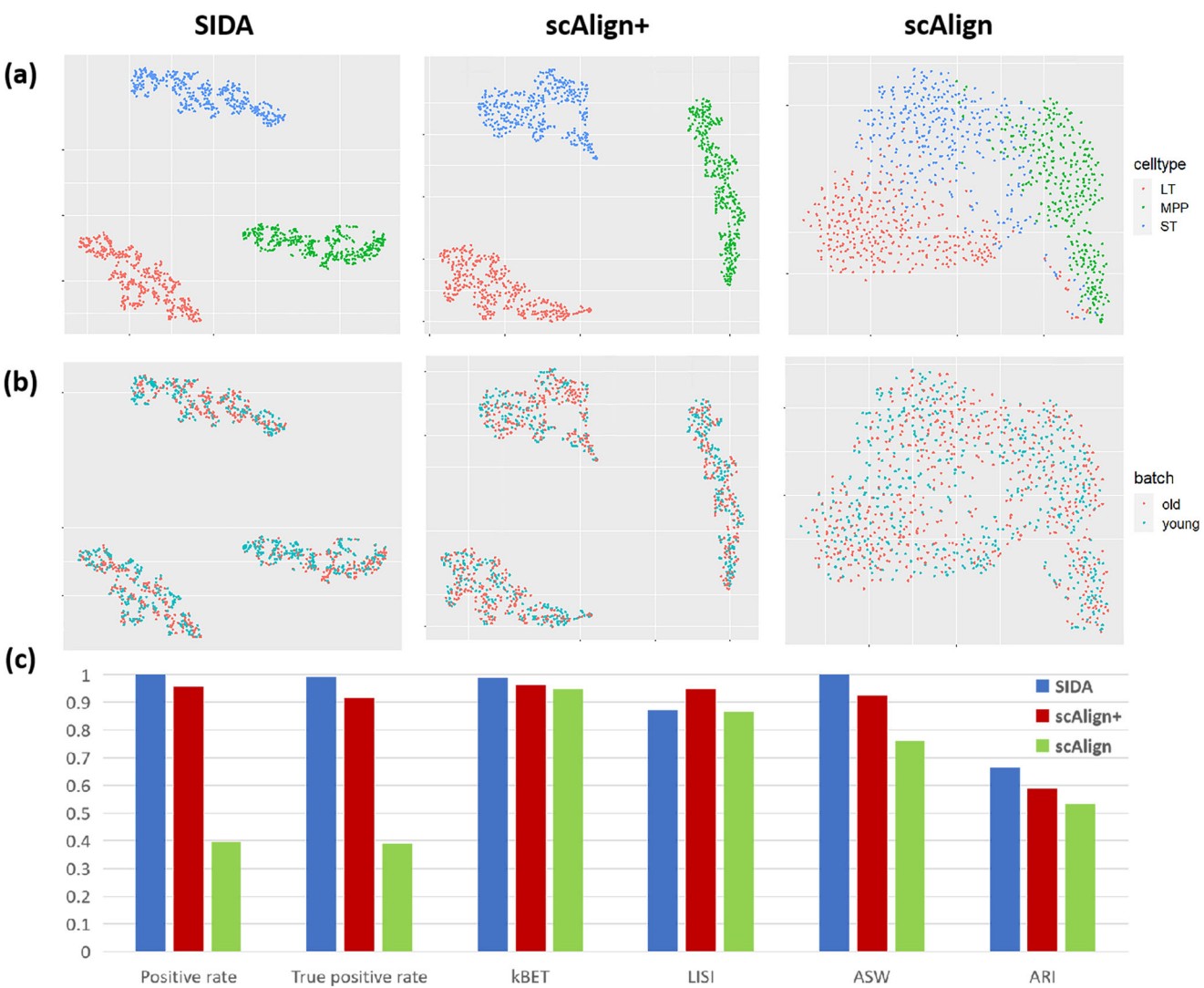

**Fig. 4 Comparing SIDA, scAlign, and scAlign+ on HSCs data collection. a** tSNE visualization of SIDA, scAlign+, and scAlign colored by cell types; **b** tSNE visualization of SIDA, scAlign+, and scAlign colored by batch labels; **c** Evaluation metrics based on the HSCs data collection.

dataset that was a poor reference by itself, integrating other datasets into this poor reference could significantly improve the performance of cell type mapping. This is consistent with the general intuition that proper data integration may lead to more comprehensive atlases that serve as better references to represent cellular distributions and heterogeneity.

Figure 6b shows the cell type mapping results in the leave-one-out analysis of the PBMC data collection. Based on the average of each row across all heatmaps in Fig. 6b, we observed that SIDA and existing supervised integration algorithms (scAlign+ and LAmbDA) showed similar performance compared to individual batches as reference, ranging between 89 and 91%, whereas the average performance of the four unsupervised integration algorithms (Seurat, Harmony, Limma, and scAlign) ranged between 89 and 90%. This result provided an example that data integration was not always necessary for cell type mapping.

Cell type mapping results in the gut data collection showed very interesting variation. As shown in the last heatmap in Fig. 6c, it seemed very challenging to predict cell types for one of the batches (Wang). When the other three batches served as the left-out query data, cell type mapping was able to achieve decent performance depending on the choice of reference. Based on the average of each row across all heatmaps in Fig. 6c, the average

performance of individual batches as reference was 67%, the average performance of SIDA integration as reference was 77%, and the average performances of the other supervised integration algorithms, scAlign+ and LAmbDA, were 67 and 74%. The average performances of the four unsupervised integration algorithms, scAlign, Seurat, Harmony, and Limma, were 68%, 63%, 60%, and 69%, respectively. As described in the previous section on integration metrics, Limma, scAlign, and scAlign+ did not mix the batches in the gut data collection, and therefore, integrated data based on these three integration methods led to similar cell type mapping accuracy compared to individual batches as reference. Although Seurat and Harmony outperformed Limma, scAlign, and scAlign+ according to the integration metrics, the batch mixing achieved by these two algorithms was at the cost of improper alignment of some of the different cell types, which negatively impacted cell type mapping accuracy when Seurat and Harmony's integrated data were used as references. This result showed that evaluations based on integration metrics and cell type mapping could provide complementary perspectives of data integration performance. In this gut data collection, SIDA integration as reference data led to an average of 77% accuracy in cell type mapping, which achieved the highest performance improvement over individual batches as reference.

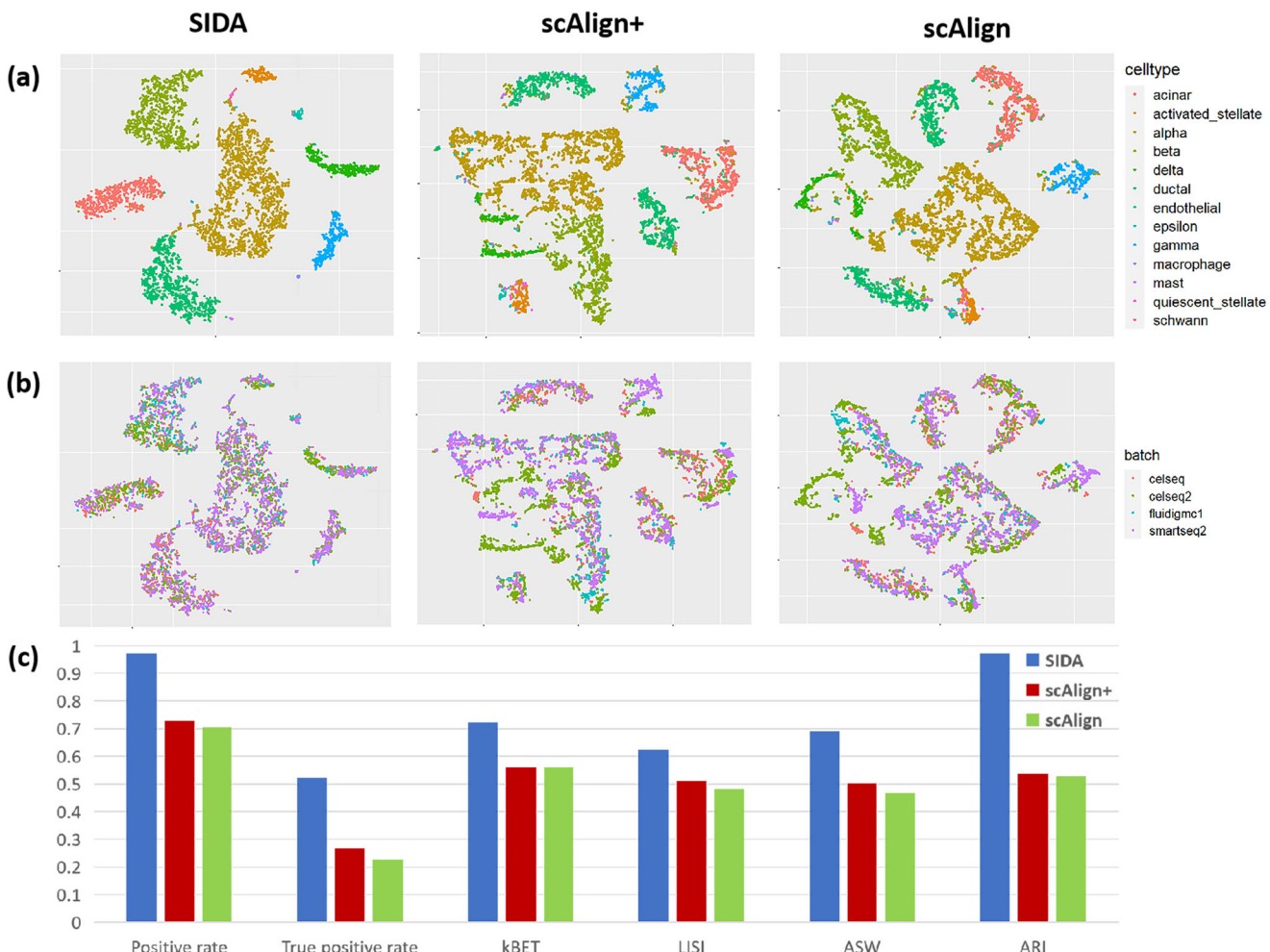

**Fig. 5 Comparing SIDA, scAlign+, and scAlign on pancreas islet data collection. a** tSNE visualization of SIDA, scAlign+, and scAlign colored by cell types; **b** tSNE visualization of SIDA, scAlign+, and scAlign colored by batch labels; **c** Evaluation metrics based on the pancreas islet data collection.

## Discussion

In this paper, we propose a supervised integration strategy for scRNA-seq data called SIDA. The key idea is to use cell type labels of individual datasets to inform the integration when cell type labels are available in the datasets to be integrated. The supervised integration is achieved using a deep neural network optimized with a Classification and Contrastive Semantic Alignment loss function to encourage the alignment of the same cell types across datasets and the separation of different cell types. When integrating scRNA-seq datasets that do not have cell type labels, SIDA is not applicable. However, when such cell type labels are available, SIDA is able to achieve better batch mixing and cell type separation, as well as improved accuracy in cell type mapping of new datasets. As global efforts of cell atlases progress, an increasing number of scRNA-seq are being accumulated, along with analysis results and cell type annotations. SIDA can be useful in any analysis that aims to summarize multiple previously analyzed datasets into larger and more comprehensive atlases.

To evaluate SIDA, we compared it with existing unsupervised and supervised integration algorithms. We applied two approaches that probed orthogonal perspectives of the integration performance. One approach was based on quantitative metrics that were previously used to benchmark unsupervised integration algorithms (i.e., positive rate, true positive rate, kBET, LISI, ASW, and ARI). These metrics were designed to quantify batch mixing and cell type separation in the embedding space. We observed

that SIDA led to improved scores in almost all metrics across pancreas, PBMC, and gut data collections. When comparing with the existing supervised integration algorithm scAlign+ on the two data collections (HSCs and pancreas islet) provided in its tutorial documentation, we observed that SIDA also led to improved scores in the majority of evaluation metrics except for LISI in the HSCs data collection. The robustness of SIDA over scAlign+ was highlighted when integrating the pancreas islet data collection, which involved multiple batches with a relatively large number of cell types. The other approach was based on performance in cell type mapping, which explicitly quantified the utility of the integrated data in cell type interpretation of new data. We observed that supervised and unsupervised integration achieved similar performance in the pancreas data collection but showed moderate to large improvements in the PBMC and gut data collections. It was encouraging that SIDA showed improved performance based on both evaluation approaches.

Implementation of one of the evaluation metrics, ARI, involves cell clustering in the integrated embedding space followed by a comparison of the resulting clusters and the known cell type labels. This clustering step requires a pre-specified definition of the number of clusters $k$. In our implementation of ARI, we set $k$ equal to the number of known cell types. This may not be the optimal choice because there is no guarantee that the $k$ resulting clusters will align with the $k$ cell types even if the integration result is perfect. However, setting $k$ larger also does not guarantee

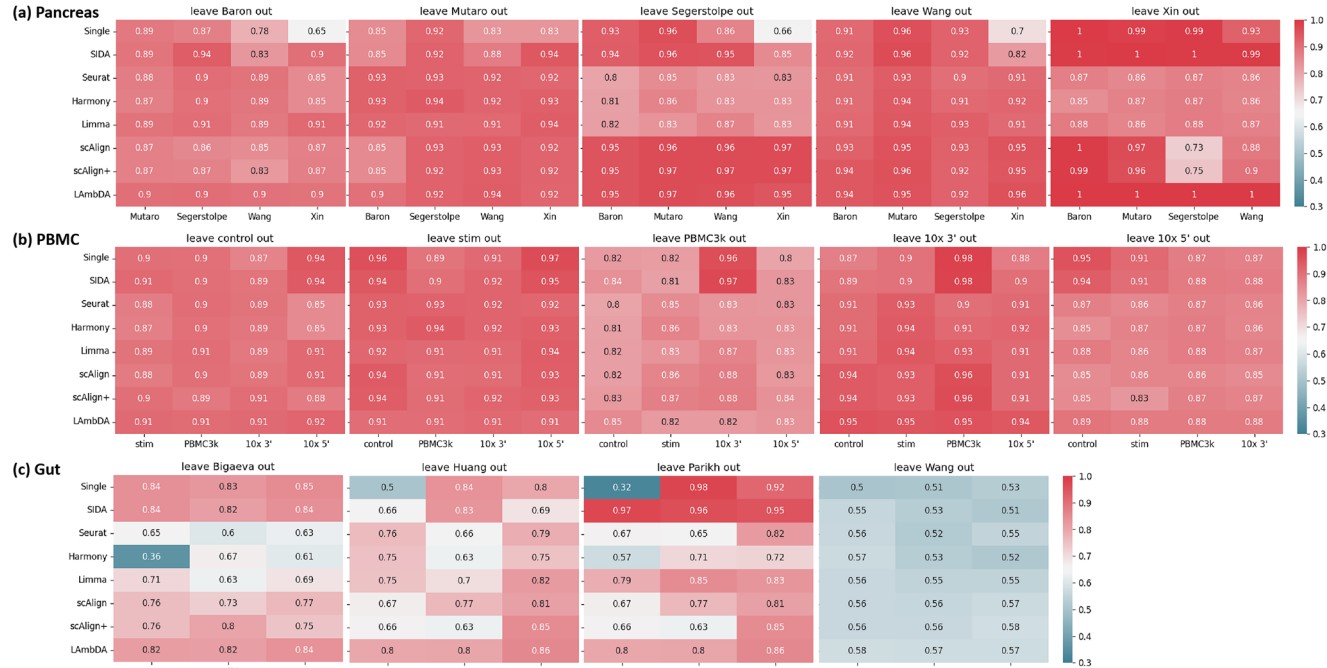

**Fig. 6 Evaluating supervised and unsupervised integration algorithms using cell type mapping and leave-one-out strategy.** Each heatmap shows the cell type mapping accuracies computed by leaving one batch out of a data collection to serve as the query data. Inside one heatmap, each element corresponds to a particular choice of reference data. Elements in the first row of a heatmap represent cell type mapping accuracies when individual batches were separately used as reference data. In the second row, the reference data were generated by SIDA results converted to different choices of target space. In the remaining rows, the reference data were generated by the existing unsupervised and supervised integration algorithms, with results converted to different choices of target space. **a** Cell type mapping accuracies in the pancreas data collection. Each heatmap corresponds to one left-out batch. **b** Cell type mapping accuracies in the PBMC data collection. **c** Cell type mapping accuracies in the gut data collection.

that the clustering results would capture all the known cell types, especially relatively rare cell types. Given the fact that ARI is sensitive to the number of clusters and penalizes over-clustering, we decided to follow the practice of ARI calculation in published benchmarking analysis for data integration, setting $k$ to be the same as the number of known cell types.

The preprocessing of SIDA involves principle component analysis to reduce the space of high variable genes down to the first 50 PCs, which serves as the input to the SIDA network. The choice of working with the PCs was largely driven by computational complexity. If the top 2000 high variable genes served as the input space, the SIDA network would include a substantially larger number of parameters, leading to significantly increased computational cost. As a separate note, when we generated the PCA space to integrate multiple batches in a data collection, we performed PCA on each batch separately. As a result, the first 50 PCs from various batches typically did not align with each other. This actually represented a more challenging situation compared to using highly variable genes where the features in different batches are the same.

The training process of SIDA involves the sampling of a subset of cells from various batches to form training pairs for SIDA to learn the differences of corresponding cell types across different batches. It is important to evaluate the robustness and reproducibility of SIDA with respect to the stochasticity involved in random sampling. We tested SIDA's consistency by applying it to the pancreas, PBMC, and gut data collections multiple times with different random seeds (see Supplementary Note 4: Evaluation of robustness and reproducibility of SIDA). In Supplementary Figs. 4 and 5, we observed low variation in the evaluation metrics and highly stable tSNE visualization of the embedding space with respect to random sampling, both indicating SIDA's robustness and reproducibility.

Since scRNA-seq data integration often aims to create comprehensive atlases that include a large number of cells, computational efficiency is an important consideration. We examined the running time of SIDA, four unsupervised algorithms (Seurat, Harmony, Limma, scAlign), and two supervised algorithms (scAlign+ and LAmbDA) across three data collections (pancreas, PBMC, and gut). The result is summarized in Supplementary Table 2 and Supplementary Note 2. Algorithms without deep learning strategy (Seurat, Harmony, and Limma) were computationally much cheaper than the other four deep-learning-based algorithms (SIDA, scAlign, scAlign+, and LAmbDA). Among all the algorithms, SIDA achieved the best integration performance and required the longest computing time. This result represents a trade-off between performance and computational cost.

Our current deep learning network for supervised integration provides integrated data in a low-dimensional embedding space, which is not able to directly serve as the reference data for cell type mapping. In order to perform cell type mapping, we apply the Mutual Nearest Neighbor strategy to convert the integrated low-dimensional embedding space to the original high-dimensional gene space, where we need to choose one of the original datasets as the target space. One future direction is to expand our deep learning network to include an encoder-decoder module which is trained to map the low-dimensional embedding space back to the high-dimensional gene space. This will lead to an end-to-end supervised integration method specifically optimized for automated cell type mapping applications.

## Methods
**Data preprocessing**. The first step of data preprocessing is to consolidate cell type annotations in the batches to be integrated because cell type annotations in different batches may have different terminologies at different levels of granularity (e.g., different abbreviations or naming conventions or different levels of details of

cell types and subtypes). To apply SIDA, we manually consolidate the cell type labels across different scRNA-seq batches to be integrated. We use the following rules to consolidate the cell type annotations. (1) We unify cell type annotations to be the most general level across the batches to be integrated. For example, if monocytes in one batch are annotated as "CD14 monocytes" or "CD16 monocytes" while monocytes in another batch are annotated just as "monocytes", we convert both "CD14 monocytes" and "CD16 monocytes" annotations in the first batch to "monocytes". (2) We unify different abbreviations and spellings of the same cell type. For example, annotations of dendritic cells may be "Dendritic Cells" in one batch, "DCs" in another batch, and "DC" in a third batch. We update the annotations so that the dendritic cells in all batches are annotated with an identical name.

The second step of data preprocessing is gene selection and dimension reduction. We filter out the nonoverlapping genes across the batches to be integrated. We then apply library size normalization and log transformation to the raw data. After that, we apply PCA to each batch respectively, and keep the first 50 PCs in each batch. The subsequent domain adaptation neural network operates in the space of the first 50 PCs instead of the space of high variable genes, which reduces the size of the neural network and makes the computational complexity tractable.

**Domain adaptation network**. We propose a deep domain adaptation neural network called SIDA to achieve supervised integration. The network architecture is shown in Fig. 1b. The preprocessed low-dimensional data (50 dimensions after PCA) is fed into the network as input. The network is composed of a Siamese network and a classification network. First, the input data is fed into the Siamese network, which has two shared-weight identical branches, "g", the first branch is for source domain data, and the second is for the target domain data. Here, the source and target domains are different batches to be integrated. "g" is a convolutional network for feature extraction, which is trained to map each batch into a common low-dimensional embedding space. To further facilitate the integration of multiple batches, a two-layer feed-forward classification network "h" is included, appended after the first branch (source domain branch). The Siamese network takes a pair of cells from two different batches for training. The two cells in the training pair are passed through the two shared-weight branches and thus are mapped into a common embedding space. As shown in Fig. 1a, training pairs are drawn from different batches in a rotated fashion. For example, if there are 3 batches to be integrated, cell pairs are generated by randomly drawing from batch 1 and 2, batch 1 and 3, batch 2 and 3, batch 2 and 1, batch 3 and 1, batch 3 and 2, and then rotating back to batch 1 and 2. Such a rotated fashion allows all batches to serve as the source domain of the Siamese network with respect to another batch as the target domain, which ensures that the network "g" is able to properly align cells from all possible pairs of batches to be integrated. Although the classification network "h" is only appended after the source domain branch, the rotated fashion of the training cell pairs enables "h" to be trained for all cell types in all batches to be integrated. This is especially important when there exist cell types that are unique to one of the batches to be integrated. When creating training cell pairs in the analyses shown in the "Results" section, we randomly selected 400 cells per cell type for each batch in the pancreas data collection, 800 cells per cell type for each batch in PBMC and gut data collections, and 250 cells per cell type for each batch in pancreas islet and HSCs data collections. We apply the Classification and Contrastive Semantic Alignment loss (Fig. 1c) to train the whole network.

The Classification and Contrastive Semantic Alignment loss function is composed of two separate loss functions: a contrastive semantic alignment loss and a classification loss. The Contrastive Semantic Alignment loss is applied to the output of network "g". The Contrastive Semantic Alignment loss function contains two components: a semantic alignment loss $L_{SA}$ and a separation loss $L_S$. Intuitively, the semantic alignment loss $L_{SA}$ minimizes the distance between cells from different batches domains with the same cell type label, which encourages the alignment of cells of the same cell type across batches. The separation loss $L_S$ maximizes the distance between cells with different cell type labels, which encourages the separation of cells of different cell types. More specifically, given two cells in a training pair from source and target batches ($X^S$ and $X^t$), if they are of the same cell type label, minimizing $L_{SA}(g) = \sum_{a=1}^{C} \frac{1}{2} \parallel g(x_a^s) - g(X_a^t) \parallel$ will encourage $X_a^S$ and $X_a^t$ to be close to each other in the embedding space, and if they are of different cell type labels, minimizing $L_S(g) = \sum_{a,b|a} \frac{1}{2} max(0, m - \parallel g(X_a^s) - g(X_b^t) \parallel)^2$ will encourage $X_a^s$ and $X_b^t$ to be far away from each other, where $C$ is the number of cell types and $m$ is the fixed margin that specifies the separability in the embedding space. The classification loss $L_C(f) = \mathbf{E}[l(f(X^s), Y)]$ is applied to train "h", which is a standard cross entropy loss. The classification-based training process further encourages the separation of different cell types and the aggregation of the same cell type, including cell types that appear in multiple batches, as well as batch-specific cell types. The output of the second feature extraction layer of "h" is the final integrated embedding space.

**Execution of pancreas, PBMC, and gut data collections**. We applied SIDA, four unsupervised integration methods (SeuratV3[21], Harmony[22], limma[23], scAlign[11]) and two supervised integration methods (scAlign+[11], LAmbDA[12]) to three data collections (pancreas, PBMC, gut), generating integrated versions for each data collection separately. The integrated datasets are evaluated in terms of both batch

mixing and cell type separation. We use a $k$-nearest neighbor-based approach to define positive rate and true positive rate, which quantify batch mixing and cell type separation[24]. We also examined evaluation metrics used in a recent benchmark paper for scRNA-seq data integration[1], including $k$-nearest neighbor batch-effect test (kBET), local inverse Simpson's index (LISI), average silhouette width (ASW), and adjusted rand index (ARI).

**Execution of pancreatic islet and HSCs data collections**. We performed an additional comparison with scAlign, which provides both unsupervised (scAlign) and supervised (scAlign+) options[11]. We performed the comparison on the pancreas islet and HSCs data collections which were used in scAlign's tutorial demonstrations (https://github.com/quon-titative-biology/scAlign). We decided to use these data collections to make sure that we properly reproduced the results in the tutorial demonstrations, which ensures a fair comparison with SIDA. For completeness, our comparison included both unsupervised and supervised options of scAlign. We use a $k$-nearest neighbor-based approach to define positive rate and true positive rate, which quantify batch mixing and cell type separation[24]. We also examined evaluation metrics used in a recent benchmark paper for scRNA-seq data integration[1], including $k$-nearest neighbor batch-effect test (kBET), local inverse Simpson's index (LISI), average silhouette width (ASW), and adjusted rand index (ARI).

**Execution of automated cell type mapping**. We applied a leave-one-out strategy to each data collection. For a given data collection, we first left out one batch and integrated the remaining batches using either SIDA or an existing integration algorithm. We then performed automated cell type mapping to predict the cell type labels of the left-out batch using the integrated data as reference.

We first converted the integrated data from the low-dimensional embedding space back to the original high-dimensional gene space. To achieve this, we picked one of the batches in the integrated data as the target space, applied the Mutual Nearest Neighbors strategy in Seurat to find anchors between the picked batch and the other batches in the low-dimensional embedding space, and used weighted differences of the anchors in the original gene space to convert the integrated low-dimensional data to the original high-dimensional gene space, so that the integrated data in high-dimensional gene space resembled the picked batch. When converting the integrated low-dimensional space to the high-dimensional gene space, we could pick any of the integrated batches as the target space; therefore, one integration algorithm produced several integrated versions of integrated data, and the number of versions was the same as the number of batches that were integrated.

After generating an integrated dataset using one integration algorithm with one choice of target space, the integrated dataset was considered as reference data for cell type mapping, and the left-out batch was considered as query data. We applied the cell mapping pipeline in scanpy[37], which first selected high variable genes and then used a PCA-based function to predict the cell type labels for the query cells based on the reference data.

**Evaluation metrics**. To evaluate the performance of the data integration, we use a $k$-nearest neighbor-based approach to quantify both batch mixing and cell type separation[24]. We also examined evaluation metrics used in a recent benchmark paper for scRNA-seq data integration[1], including $k$-nearest neighbor batch-effect test (kBET), local inverse Simpson's index (LISI), average silhouette width (ASW), and adjusted rand index (ARI).

To quantify both batch mixing and cell type separation, we used the metric in ref. [24] based on the $k$-nearest neighbors (kNNs) of cells. First, we classify all cells into 'positive' and 'negative' cells. 'Positive' cells are those surrounded mostly by cells from the same cell type. In our analysis, one cell is classified as 'positive' if at least 95% of its $k$-nearest neighbors are of the same cell type, and $k$ is set as 50. Then, the 'positive' cells are further classified into 'true positive' and 'false positive' cells. 'True positive' cells are those surrounded by appropriate proportions of cells from different batches. A 'positive' cell is classified as 'true positive' if the batch distribution of its neighborhood is consistent with the global batch distribution. The three-sigma rule is used to measure the consistency of distribution. For a 'positive cell' of a certain cell type, assume the number of cells of this cell type in the $n$ batches are $N_1, N_2, ..., N_n$, and therefore, the distribution of this cell type's cells across the batches is $p_i = \frac{N_i}{\sum_{j=1}^{n} N_j}, i = 1, 2, ..., n$. For the 'positive cell', we focus on its $k = 50$ neighbors and denote the number of neighbors from the batches as $M_1, M_2, ..., M_n$. If the batches are well mixed and integrated, we expect the distribution of $M_i$ to be within three standard deviations around the distribution of $p_i$. More specifically, $m_i$ should be in the range of $[max(0, kp_i - 3\sqrt{kp_i(1 - p_i)}), kp_i + 3\sqrt{kp_i(1 - p_i)}]$ for all $i = 1, 2, ..., n$. The percentage of 'positive' cells and the percentage of 'true positive' cells serve as metrics to quantify integration performance.

kBET measures batch mixing at the local level, which compares the kNN local distribution against global distribution using Pearson's $\chi^2$ test. First, a $k$-nearest neighbor graph is constructed based on the integrated embedding space. Then, 10% of the cells are chosen, and the batch distribution of the nearest neighbors of each chosen cell is compared with the global distribution of the batches using the $\chi^2$-test. If the local distribution is sufficiently similar to the global distribution, the $\chi^2$ test does not reject the null hypothesis that there is a good batch mixture around the chosen cell. The rejection rate ranges from 0 to 1. Here, we use (1-rejection rate) as

the final kBET value, and a kBET value close to 1 signifies the batches are well mixed.

LISI measures the effective diversity of local distributions, which can be applied to quantify both cell type separation and batch mixing. First, LISI selects the nearest neighbors based on the local distribution of distance with a fixed perplexity. Then, it computes the inverse Simpson's index for the diversity of selected neighbors, which reflects how many different types are in a neighborhood and how evenly distributed the population of each type is. For a given neighborhood, the formula to calculate inverse Simpson's index is $1/\sum_{b=1}^{B} p(b)$. The probabilities $p(b)$, $b = 1, 2, …, B$ here refer to the batch probabilities in the local neighborhood distributions described above. When the type in LISI is defined by batch, the resulting score (iLISI) quantifies batch mixing, and a higher iLISI value indicates better batch mixing. When the type in LISI is defined by cell type, the resulting score (cLISI) quantifies cell type separation, and a lower cLISI value indicates better cell type separation. The harmonic mean of cLISI and iLISI is computed to combine the evaluations for cell type separation and batch mixing into an overall score $F_{1_{LISI}} = \frac{2(1-\text{cLISI})(\text{iLISI})}{1-\text{cLISI}+\text{iLISI}}$.

ASW uses the average silhouette score to quantify cell type separation and batch mixing. For one data point, its silhouette score is computed by subtracting its average distance to other members in the same cluster from its average distance to all members of the nearest neighboring cluster and then dividing by the larger of the two values. The resulting score ranges from −1 to 1, where a higher value indicates that the data point fits well in its cluster. When the distances are computed in the integrated embedding space, and the clusters are defined by cell types, the ASW is denoted as $\text{ASW}_{\text{celltype}}$, with a higher value indicating cell clusters are well separated in the embedding space. When the distances are computed in the integrated embedding space, and the clusters are defined by batch labels, the ASW is denoted as $\text{ASW}_{\text{batch}}$, with a lower value indicating batches are well mixed in the embedding space. The harmonic mean of the two ASW values is used to combine them into an overall score: $F_{1_{ASW}} = \frac{2(1-\text{ASW}_{\text{batch}})(\text{ASW}_{\text{celltype}})}{1-\text{ASW}_{\text{batch}}+\text{ASW}_{\text{celltype}}}$.

ARI measures the agreement between two sets of cluster labels, which can be applied to quantify both cell type separation and batch mixing. First, $k$-means is applied to cluster cells in the integrated embedding space and generates predicted clustering labels, where $k$ is the number of unique cell types in the batches to be integrated. Then, the ARI between the $k$-means predicted cluster labels and the true cell type labels is calculated and denoted as $\text{ARI}_{\text{celltype}}$, where a higher value corresponds to better cell type separation. A second ARI value between the $k$-means predicted cluster labels and the batch labels is calculated and denoted as $\text{ARI}_{\text{batch}}$, where a lower value corresponds to better batch mixing. The harmonic mean of the two ARI values is used to combine the two aspects into an overall score $F_{1_{ARI}} = \frac{2(1-\text{ARI}_{\text{batch}})(\text{ARI}_{\text{celltype}})}{1-\text{ARI}_{\text{batch}}+\text{ARI}_{\text{celltype}}}$.

**Statistics and reproducibility**. The statistical tests used in this study were performed using R 4.2.1 or Python 3.7, and details of statistical analyses are described in the "Methods" section. We have provided the reproducibility evaluation in Supplementary Note 4: Evaluation of robustness and reproducibility of SIDA.

**Reporting summary**. Further information on research design is available in the Nature Portfolio Reporting Summary linked to this article.

## Data availability
The references of data that support the findings of this study are provided in Supplementary Table 1.

## Code availability
SIDA code is available at https://github.com/syt960909/SIDA. This code is implemented in Python, and all the required packages are listed in requirement.txt in the GitHub repository.

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

## Acknowledgements

This publication is part of the Gut Cell Atlas Crohn's Disease Consortium funded by The Leona M. and Harry B. Helmsley Charitable Trust and is supported by a grant from Helmsley to Georgia Institute of Technology (www.helmsleytrust.org/gut-cell-atlas/). This work was also supported by the National Science Foundation (CCF2007029). P.Q. is an ISAC Marylou Ingram Scholar and a Wallace H. Coulter Distinguished Faculty Fellow. The funders had no role in study design, data collection and analysis, the decision to publish, or the preparation of the manuscript.

## Author contributions

Y.S. conceived the study and conducted the experiments. P.Q. provided overall guidance and support; Y.S. and P.Q. drafted and revised the manuscript.

## Competing interests

The authors declare no competing interests.
