## [Peer Review File · Communications Biology]

Reviewers' comments:

Reviewer #1 (Remarks to the Author):

In this manuscript, Sun and Qiu present a new supervised integration method of multiple scRNA-seq datasets. Their method adopts domain adaptation deep neural network to integrate scRNA-seq data with the help from cell type annotation. Compared to unsupervised methods, their method can give more accurate integration, as well as improve the accuracy of cell type mapping analysis. However, there are still several concerns must be addressed before further consideration.

Major comments:

1. Authors mentioned in the Introduction that scAlign is the only method for supervised integration, and they only compared their method to scAlign and scAlign+. However, there are several other supervised methods published in the recent two years, for example, SMNN and iSMNN. Both methods have already showed to outperform unsupervised strategies in data integration, such as Seurat and Harmony, which were also compared by authors in this study. So it is worthy to compare the domain adaptation method to these published supervised methods, to provide a more solid evidence to its advantages.
2. Authors showed that domain adaptation method outperform Seurat and Harmony in cell type mapping analysis. Why not compare to scAlign and scAlign+ as well, especially scAlign+ is an integration method?
3. It is worthy to show the computational efficiency of domain adaptation method, such as computing time and memory consuming, for the analysis. The sources required by the computing is important for users to make a decision on which method they want to use.
4. Is the domain adaptation method sensitive to the number of cell types shared across batches? For example, does it still work well if only one cell type shared between batch 1 and 2, while another cell type shared between batch 2 and 3?
5. All the data are visualized by t-SNE. Why not UMAP?

Minor comments:

1. The fifth paragraph of Introduction and the first section of Results are similar. It is better to move some contents to Results and make it more brief in Introduction.
2. In the Introduction, authors said Seurat is CCA-based strategy, however, although using CCA, the core part of Seurat3 is still MNN based. Seurat2 is purely based on CCA.
3. There are some small typos in the manuscript. Please go over the entire manuscript carefully to make sure all are corrected.
4. The original references should be provided to all the datasets used in method evaluation.
5. Some descriptions in the main text should be moved to figure legend.

Reviewer #2 (Remarks to the Author):

The authors propose a new supervised single cell dataset integration approach to reduce batch effects. This is already a very heavily studied mature field in single cell data analytics. This usually makes me skeptical about new methodological contributions because so many approaches have already been proposed increasing the threshold for novel contribution. However, the authors perform many baseline comparisons, on many datasets, evaluated using many evaluation metrics and perform well against other approaches. Since there does appear to be some advantage to the new approach, I would suggest more clearly stating the novel methodological contribution of the method that makes it perform better. I have a number of major concerns which if addressed would make this manuscript suitable for publication.

Major concerns:

Though not the current state-of-the-art, there have been other approaches that utilize aspects of supervised learning during batch correction, such as LAMBDA (PMID: 31038689) and SMNN (PMID: 32591778). The LAMBDA method also has losses similar to the semantic alignment loss and the separation loss. There should be some discussion of these previous approaches as they relate to the current approach. What aspects of the approaches are shared and how has this new method made improvements?

The code needs to be more readily usable. I would suggest making this into a python executable with clearly defined input files that can be run from command line. There should also be more clear documentation and a github site. The ipynb is good for demonstration purposes but is not very useful if someone were to attempt to use this software for their own purposes. There should also be vignettes on the github site showing various examples of how to use this method. There should also be clear documentation about what dependencies are necessary, python version, ML library/version, python libraries, and compatibility with operating systems.

The figures are missing a lot of information. Please label all axes in all figure panels. For instance, the scatter plots are tSNE but I don't know since the axes are not labeled. Also, in the bar plot panels please label the y-axis. The tick mark labels on t-SNE are too small to read. Why did the authors use tSNE instead of UMAP?

I would suggest giving this algorithm a name. There are already a few supervised integration methods available for single cell RNA sequencing so it does not make sense to call this one "supervised integration" since that describes a whole set of available algorithms.

Please list the evaluate metrics in a table(s) with the best values marked in bold/italicized. It is important to have a benchmark table, preferably with variability in metrics across multiple iterations, so that readers can quickly see the differences in performance numerically.

There should be an experiment to highlight the consistency of the results. I would suggest performing an experiment multiple times using different starting seeds to show how much variability is in the results from the methods.

The methods section is not detailed enough. It needs to be expanded to clearly explain how every experiment was performed. It currently only briefly describes the datasets, domain adaptation, and evaluation metrics. One simple way to organize this is to have corresponding sections in the methods to those in the results that explain how the experiments were conducted.

Minor concerns:

Why were principal components used as input instead of genes? Is there support in the literature for this?

If there are k cell types, why should there by definition be k clusters? Is it possible that there would be subtypes of cells making $>k$ clusters?

Response to Review

"Domain adaptation for supervised integration of scRNA-seq data"

Yutong Sun, Peng Qiu

We are grateful for the reviewers' comments and suggestions. Below, we first include a summary of the changes we have made during the revision. After that, we include the review comments and provide our point-by-point responses. Thank you for your time in reviewing this work.

Description of the changes made:

Per suggestions from the editor, we 1) further performed comparisons to other supervised integration methods, 2) gave the proposed algorithm a specific name, SIDA, 3) created a GitHub repository for code sharing, 4) made updates to the figures, 5) removed some of the overlapping contents between Introduction and Results sections, 6) expanded the Method section, 7) added experiments on computational efficiency, and 8) showed the consistency of the proposed algorithm.

Reviewer 1's comments:

Major comments:

1. Authors mentioned in the Introduction that scAlign is the only method for supervised integration, and they only compared their method to scAlign and scAlign+. However, there are several other supervised methods published in the recent two years, for example, SMNN and iSMNN. Both methods have already showed to outperform unsupervised strategies in data integration, such as Seurat and Harmony, which were also compared by authors in this study. So it is worthy to compare the domain adaptation method to these published supervised methods, to provide a more solid evidence to its advantages.

We would like to thank the reviewer for the insightful comments. We agree with the reviewer's point that there are several other supervised integration methods, and it is interesting to compare with them. In addition to scAlign+, we also included SMNN, iSMNN and LAMBDA in the introduction, which are all supervised integration algorithms. We added the comparison with scAlign+ and LambDA in subsections 3 and 5 in the Results section. However, we were unable to include SMNN and iSMNN. On the GitHub repo for SMNN and iSMNN, although their authors provided both code and a well-documented tutorial, when we tried to run their example code on their example data, we encountered errors. We noticed on GitHub that others have experienced the same problem and posted/asked questions about the errors, but those errors have not been resolved. We have contacted the authors of SMNN and iSMNN to ask about these errors, but have not received any feedback. Therefore, we were unable to include SMNN and iSMNN in our comparison.

2. Authors showed that domain adaptation method outperform Seurat and Harmony in cell type mapping analysis. Why not compare to scAlign and scAlign+ as well, especially scAlign+ is an integration method?

We have added the comparisons. Please refer to the response to the first major comment

3. It is worthy to show the computational efficiency of domain adaptation method, such as computing time and memory consuming, for the analysis. The sources required by the computing is important for users to make a decision on which method they want to use

We agree with the reviewer that it would be helpful to compare the computational efficiency of our approach and existing data integration algorithms, which we summarized in the table below. This table shows that algorithms without using deep learning strategy (Seurat, Harmony and Limma) are computationally much cheaper than the four deep-learning-based algorithms (SIDA, scAlign, scAlign+, and LAMBDA). Among all the algorithms, SIDA achieves best integration performance and has the longest computing time. This table is included in the supplementary materials as Table S2.

This result represents a trade-off between performance and computational cost. In terms of how to decide which integration method should be used, our opinion is that:

- when integrating newly generated datasets without prior analysis and cell type labels, we have to use unsupervised integration algorithms, such as Seurat and Harmony.
- when integrating previously analyzed datasets with cell type labels available, we believe it is better to use supervised integration, and we would prefer our method, SIDA. Although Table S2 shows that SIDA requires longer computing time compared to other deep-learning-based supervised integration algorithms, we have demonstrated in Table 1 and Figure 3c that SIDA significantly outperformed the other methods in the gut data collection, which is the most challenging data collection analyzed in this study. The table below is included in the supplementary material.

Table. Computation efficiency of SIDA, four unsupervised algorithms (Seurat, Harmony, Limma, scAlign), and two supervised algorithms (scAlign+ and LAMBDA) applied to three data collections.

		SIDA	Seurat	Harmony	Limma	scAlign	scAlign+	LAMBDA
pancreas	Computing time	8hours	9mins	49S	1MIN	7.5hours	6hours	9hours
	Memory consuming	2632MB RAM +3865MB GPU*	11562.7MB RAM	13949.9MB RAM	13433.6MB RAM	10024.1MB RAM	10011.7MB RAM	954MB RAM +3875MB
PBMC	Computing time	28hours	32mins	2mins	3mins	6hours	5hours	12hours
	Memory consuming	797.9MB RAM +4149MB GPU	29454.4MB RAM	23893.3MB RAM	32903.1MB RAM	20457MB RAM	27870MB RAM	983.7MB RAM +3822MB GPU
gut	Computing time	25hours	20mins	1min28s	1min15s	2hours	3.3hours	11hours
	Memory consuming	1098.9MB RAM +3945MB GPU	21552.6MB RAM	17746.2MB RAM	17105.5MB RAM	47004.7MB RAM	47063.4MB RAM	1243.4MB RAM +3862MB GPU

*The GPU used here is NVIDIA Quadro P2200.

4. Is the domain adaptation method sensitive to the number of cell types shared across batches? For example, does it still work well if only one cell type is shared between batch 1 and 2, while another cell type is shared between batch 2 and 3?

This is an interesting question. For unsupervised integration algorithms that aim to align the distributions of different batches, the number of shared cell types across batches cannot be too small. In contrast, supervised integration algorithms should still work well even if there is only one shared cell type between batches, because the known cell type labels can guide supervised integration to only force the shared cell types to overlap in the integrated embedding space.

Even though we do not have such a scenario in our experiments, we believe SIDA is able to produce a fair integration when there is only one cell type shared between batch 1 and 2, and another cell type shared between batch 2 and 3. In our algorithm, we first generate cell pairs for training the SIDA model. Each cell pair includes two cells, from two different batches. The cell types of these two cells can either be the same, or be different. Our Classification and Contrastive Semantic Alignment loss encourages the alignment of the same cell types and the separation of the different cell types. For the given scenario in this comment, assume that we have three batches: batch 1 contains cell types A, B, C; batch 2 contains cell types C, D, E; batch 3 contains cell types E, F, G. Since cell type labels are part of the input required by SIDA (or supervised integration in general), the algorithm knows batch 1 cells of cell type C should overlap with batch 2 cells of cell type C, and the cells belonging to cell types A, B, D, E should be separate from cell type C and separate from each other. Similar argument can be made for the cell types in batches 2 and 3. Conceptually, supervised integration is able to handle all kinds of scenarios regarding shared cell types between batches, including when the number of shared cell types across batches is very small.

5. All the data are visualized by t-SNE. Why not UMAP

Regarding the question of why we used tSNE instead of UMAP, this was actually an arbitrary choice. In our own analysis, we have applied both tSNE and UMAP to visualize the embedding space generated by various data integration algorithms, and the interpretations of both visualization tools are highly consistent with each other. Therefore, we arbitrarily picked tSNE to visualize the embedding space in our initial submission. In this revision, to address this question of why not UMAP, we now provide UMAP visualizations in supplementary Figures S1, S2 and S3, which correspond to the tSNE visualizations in Figures 2, 4 and 5 in the main text. Since the interpretations of these UMAP and tSNE visualizations are the same, we decided to only show tSNE visualizations in the main text, and include the UMAP visualizations in the supplementary materials.

Minor concerns and suggestions:

1. The fifth paragraph of Introduction and the first section of Results are similar. It is better to move some contents to Results and make it more brief in Introduction.

Thanks for this comment. We agree with the reviewer that there are some duplicated contents. We have removed some contents from Introduction to Results in the submitted revision.

2. In the Introduction, authors said Seurat is CCA-based strategy, however, although using CCA, the core part of Seurat3 is still MNN based. Seurat2 is purely based on CCA

Thanks for the comment. Although we only cite Seurat2 for using CCA, we agree that the statement of Seurat using CCA is ambiguous. Hence, we have changed Seurat to SeuratV2 in the submitted revision.

3. There are some small typos in the manuscript. Please go over the entire manuscript carefully to make sure all are corrected.

We have gone through the manuscript and corrected the typos.

4. The original references should be provided to all the datasets used in method evaluation

Thanks for the comment. We agree with the reviewer that the original references for all individual datasets should be provided. In this revision, we have listed the original references in the table attached below, and this table is included in the supplementary materials as Table S1.

Data collection	Batch	Reference
Pancreas	Baron	Baron M, Veres A, Wolock S L, et al. A single-cell transcriptomic map of the human and mouse pancreas reveals inter-and intra-cell population structure[J]. Cell systems, 2016, 3(4): 346-360. e4.
	Muraro	Muraro MJ, Dharmadhikari G, Grun D, Groen N, Dielen T, Jansen E, et al. A single-cell transcriptome atlas of the human pancreas. Cell Syst. 2016;3:385–394.e3.
	Segerstolpe	Segerstolpe A, Palasantza A, Eliasson P, Andersson E-M, Andreasson A-C, Sun X, et al. Single-cell transcriptome profiling of human pancreatic islets in health and type 2 diabetes. Cell Metab. 2016;24:593–607.
	Wang	Wang YJ, Schug J, Won K-J, Liu C, Naji A, Avrahami D, et al. Single-cell transcriptomics of the human endocrine pancreas. Diabetes. 2016;65:3028–38.
	Xin	Xin Y, Kim J, Okamoto H, Ni M, Wei Y, Adler C, et al. RNA sequencing of single human islet cells reveals type 2 diabetes genes. Cell Metab. 2016;24:608–15.
PBMC	Control	Butler A, Hoffman P, Smibert P, et al. Integrating single-cell transcriptomic data across different conditions, technologies, and species[J]. Nature biotechnology, 2018, 36(5): 411-420.
	Stim	
	PBMC3k	Satija Lab (2020). pbmc3k.SeuratData: 3k PBMCs from 10X Genomics. R package version 3.1.4.
	10x 3'	Zheng GXY, Terry JM, Belgrader P, Ryvkin P, Bent ZW, Wilson R, et al. Massively parallel digital transcriptional profiling of single cells. Nat Commun. 2017;8:14049.
Gut	10x 5'	
	Bigaeva	Bigaeva, E., Uniken Venema, W. T., Weersma, R. K. & Festen, E. A. Understanding human gut diseases at single-cell resolution. Hum. Mol. Genet. 29, R51–R58 (2020).
	Huang	Huang, B. et al. Mucosal profiling of pediatric-onset colitis and ibd reveals common pathogenics and therapeutic pathways. Cell 179, 1160–1176 (2019).
	Parikh	Parikh, K. et al. Colonic epithelial cell diversity in health and inflammatory bowel disease. Nature 567, 49–55 (2019).
pancreatic islet	Wang	Wang, Y. et al. Single-cell transcriptome analysis reveals differential nutrient absorption functions in human intestine. J. Exp. Medicine 217 (2020).
	CEL-Seq	Muraro M J, Dharmadhikari G, Grün D, et al. A single-cell transcriptome atlas of the human pancreas[J]. Cell systems, 2016, 3(4): 385-394. e3.
	CEL-Seq2	
	Fluidigm C1	
HSCs	Smart-Seq2	
	old	Kowalczyk M S, Tirosh I, Heckl D, et al. Single-cell RNA-seq reveals changes in cell cycle and differentiation programs upon aging of hematopoietic stem cells[J]. Genome research, 2015, 25(12): 1860-1872.
	young	

5. Some descriptions in the main text should be moved to figure legend.

We went through the manuscript, and noticed that this comment may be referring to Figure 6, which shows the evaluation of supervised and unsupervised integration algorithms using cell type mapping and leave-one-out strategy. The caption of this figure would benefit from more detailed descriptions. Following the reviewer's comment, we have removed some of the relevant descriptions in the Result section, and moved those descriptions into the caption of Figure 6.

Reviewer 2's comments:

Major concerns and suggestions:

1. Though not the current state-of-the-art, there have been other approaches that utilize aspects of supervised learning during batch correction, such as LAMBDA (PMID: 31038689) and SMNN (PMID: 32591778). The LAMBDA method also has losses similar to the semantic alignment loss and the separation loss. There should be some discussion of these previous approaches as they relate to the current approach. What aspects of the approaches are shared and how has this new method made improvements?

We would like to thank the reviewer for his/her time and efforts in reviewing our manuscript. Reviewer #1 also pointed out these algorithms. We agree with the reviewer's comment that there are several other supervised methods, which should be discussed or compared. In addition to scAlign+, we have included SMNN, iSMNN and LAMBDA in the Introduction section. In the Results section, we added comparisons with scAlign+ and Lambda in the 3rd and 5th subsections, where we compared these algorithms in terms both their performance evaluation in the integrated embedding space and their impact to accuracy of automated cell type mapping. However, we were unable to include SMNN and iSMNN in the comparisons. On the GitHub repo for SMNN and iSMNN, although their authors provided both code and a well-documented tutorial, when we tried to run their example code on their example data, we encountered errors. We noticed on GitHub that others have experienced the same problem and posted/asked questions about the errors, but those errors have not been resolved. We have contacted the authors of SMNN and iSMNN to ask about these errors, but have not received any feedback. Therefore, we were unable to include SMNN and iSMNN in our comparison.

2. The code needs to be more readily usable. I would suggest making this into a python executable with clearly defined input files that can be run from command line. There should also be more clear documentation and a github site. The ipynb is good for demonstration purposes but is not very useful if someone were to attempt to use this software for their own purposes. There should also be vignettes on the github site showing various examples of how to use this method. There should also be clear documentation about what dependencies are necessary, python version, ML library/version, python libraries, and compatibility with operating systems.

Thanks for the comment. We agree that creating a GitHub site makes it more convenient to use. Our project can now be found at <https://github.com/syt960909/SIDA>.

3. The figures are missing a lot of information. Please label all axes in all figure panels. For instance, the scatter plots are tSNE but I don't know since the axes are not labeled. Also, in the bar plot panels please label the y-axis. The tick mark labels on t-SNE are too small to read. Why did the authors use tSNE instead of UMAP?

We appreciate the reviewer for pointing this out. We agree that the tSNE map axes are unclear and misleading. Since tSNE is a nonlinear dimension reduction tool to visualize high dimensional distributions in two-dimensional space, the axes of tSNE plots actually do not carry interpretable meaning. Therefore, to avoid misleading interpretation, we followed a benchmark paper for scRNA-seq data integration (<https://genomebiology.biomedcentral.com/articles/10.1186/s13059-019-1850-9>) and removed the axis labels of tSNE plots.

We agree that readers may wonder why we choose to use tSNE instead of UMAP. Reviewer#1 also asked this exact same question. In our own analysis, we have applied both tSNE and UMAP to visualize the embedding space generated by various data integration algorithms, and the interpretations of both visualization tools are highly consistent with each other. Therefore, we arbitrarily picked tSNE to visualize the embedding space in our initial submission. In this revision, to address this question of why not UMAP, we now provide UMAP visualizations in supplementary Figures S1, S2 and S3, which correspond to the tSNE visualizations in Figures 2, 4 and 5 in the main text. Since the interpretations of these UMAP and tSNE visualizations are the same, we decided to only show tSNE visualizations in the main text, and include the UMAP visualizations in the supplementary materials.

4. I would suggest giving this algorithm a name. There are already a few supervised integration methods available for single cell RNA sequencing so it does not make sense to call this one “supervised integration” since that describes a whole set of available algorithms.

We agree with this suggestion and we have named this algorithm SIDA (**S**upervised **I**ntegration using **D**omain **A**daptation).

5. Please list the evaluate metrics in a table(s) with the best values marked in bold/italicized. It is important to have a benchmark table, preferably with variability in metrics across multiple iterations, so that readers can quickly see the differences in performance numerically

Thanks for the suggestion. We have added such a table in the revision, the new Table 1 in the main text.

6. There should be an experiment to highlight the consistency of the results. I would suggest performing an experiment multiple times using different starting seeds to show how much variability is in the results from the methods.

Thanks for the comments. We agree that the consistency and robustness of SIDA should be examined. Since the SIDA model training requires sampling of a subset of cells from each cell type and each batch, it is important to evaluate the variability of SIDA with respect to the random sampling. Following this suggestion, we performed a consistency evaluation experiment on the pancreas, PBMC and gut data collections. For each data collection, we trained three SIDA models using cells sampled with different random seeds. We

used the 6 metrics (true positive rate, positive rate, kBET, LISI, ASW, ARI) and tSNE visualization to examine the trained models. If the metrics are similar across the SIDA models constructed from the same data collection, we can be more confident in the robustness and reproducibility of SIDA. Results of this analysis is included in the supplementary materials.

The evaluations of the 6 metrics across the three data collections are summarized in Figure S4, where each panel includes the evaluation on one data collection. The three colors represent the three models trained based on different random seeds. For majority of the metrics in the three data collections (each group of bars with the three colors), SIDA models achieved low variation, lower than the variations across metrics within one data collection, especially for the metrics dominated by cell type purity. Therefore, this result demonstrated SIDA's robustness and reproducibility.

Supplementary Figure S5 shows the tSNE visualization of the embedding space generated by the three SIDA models trained by randomly sampled cells from the three data collections. In each section of Figure S5, we can observe that all the three SIDA models achieve highly similar tSNE visualization of the embedding space, except for some rotations and rearrangements of clusters which tSNE tends to do. Therefore, consistency of tSNE visualizations of the embedding space further supported SIDA's robustness and reproducibility.

7. The methods section is not detailed enough. It needs to be expanded to clearly explain how every experiment was performed. It currently only briefly describes the datasets, domain adaptation, and evaluation metrics. One simple way to organize this is to have corresponding sections in the methods to those in the results that explain how the experiments were conducted.

Thanks for the comment. We have updated the methods section to include more details and explanations of how the experiments were conducted.

Minor concerns and suggestions:

1. Why were principal components used as input instead of genes? Is there support in the literature for this?

In our implementation of SIDA, we used the first 50 PCs for each batch in a data collection to serve as input. If we choose to use high variable genes, for example, top 2000, the dimensionality of the input layer of our domain adaptation network will be a lot larger, leading to significantly increased computational cost. Therefore, in order to limit the computational complexity, we chose to work with the PCs, instead of the highly variable genes.

As a separate note, when we generated the PCA space to integrate multiple batches in a data collection, we performed PCA on each batch separately. As a result, the top 50 PCs from batch 1 typically are not the same as the top 50 PCs from batch 2, etc. This actually represents a more challenging situation compared to using highly variable genes, where the features/genes in different batches are the same.

2. If there are k cell types, why should there by definition be k clusters? Is it possible that there would be subtypes of cells making $>k$ clusters?

Within this manuscript, the only analysis that requires clustering is the ARI metric (one of the 6 evaluation metrics). To compute ARI for an integrated embedding space, we need to first cluster the cells and then compare the clustering result with the known cell type labels using ARI. This process requires a pre-specified definition of number of clusters, and we set the number of clusters to be the number of known cell types. Therefore, we believe this comment is about how to set appropriate k in the ARI calculation.

We agree with the reviewer that setting k equal to the number of known cell types may not be the optimal choice for ARI calculation, because there is no guarantee that the k resulting clusters will align with the k cell types even if the integration result is perfect. It is indeed possible that some of the k clusters may be driven by subtypes of cells, pointing to the question of whether the number of clusters should be larger than the number of known cell types. However, there are three reasons against this idea.

- (1) Setting k larger also does not guarantee that the clustering results would capture all the known cell types, especially relatively rare cell types.
- (2) ARI is sensitive to the number of clusters. If we over-cluster the data ($k >$ number of cell types), even if the clusters perfectly capture all known cell types and some of the subtypes, the ARI values against the known cell types is not going to be great, because ARI penalizes over-clustering
- (3) ARI implementations in published benchmarking studies typically choose k to be the same as the number of known cell types, maybe due to the two reasons above

Therefore, we decided to follow the typical practice of ARI evaluation for integration results, setting k to be the same as the number of known cell types in the cell type annotations of the batches.

In addition to ARI, we have 5 other performance evaluation metrics to compare various integration algorithms. Since all of the 6 performance evaluation metrics showed that SIDA achieves the overall best performance, we feel that our implementation of ARI (setting k to be the number of known cell types) provided a reasonable comparison of the algorithms.

Reviewers' comments:

Reviewer #1 (Remarks to the Author):

All my concerns have been well addressed

Reviewer #2 (Remarks to the Author):

Reviewer 2

First of all, I really appreciate that the authors put in the effort for all of the additional experiments. The authors have made many significant improvements to their manuscript. I have a few more straightforward but necessary concerns that need to be addressed before publication.

Major concerns

1) The authors have addressed my concerns.

2) I appreciate the improved documentation and github site. However, the train.py function is still not that simple to use. I would suggest making the script able to run from the command line such that ./train.py input_file1.txt input_file2.txt option1 option2 etc. Also, the readme should include an example of running the script on the example data. If the authors do not want to convert into an R or python package then there should be a clear command that can be run with clear documentation of the input files, required file formats, and various options that can be set when running train.py. All of the dependencies, including python packages, should be listed in the README file as well.

3) Please be sure to include this reasoning for omitting the tSNE tick labels and cite Tran et al. in that sentence. This explanation about UMAP is satisfactory.

4) The authors have addressed my concerns.

5) The authors have addressed my concerns.

6) The authors have addressed my concerns.

7) I see in the supplementary information there are additional explanations of the experiments. However, these are never cited in the text. If the authors do not want to move more of this text into the main text, then it should be cited in the main text where appropriate. For example: It was also important to evaluate the robustness of our algorithm to different initializations (see Supplementary Materials: Evaluation of robustness and reproducibility of SIDA).

Minor concerns

1) Please include this reasoning for using PCs instead of gene features somewhere in the main text methods Data preprocessing section. Please also comment on using PCs over genes in the discussion.

2) Please include in the discussion some comments about how k was set to the number of cell types because this manuscript is primarily concerned with baseline comparisons. Mention also that depending on the data there may be more subtypes of cells resulting in a greater number of clusters.

Response to Review

"Domain adaptation for supervised integration of scRNA-seq data"

Yutong Sun, Peng Qiu

We are glad that both reviewers think we have addressed most of the concerns raised by them. In this second round of revision, we have made a couple of further changes. Below, we first include a summary of the changes we have made. We also include the reviewers' comments and provide our point-by-point responses. Thank you for your time in reviewing this work.

Summary the changes made:

In this revision, we have (1) added information and discussions in the main text according to reviewer 2's suggestions; (2) improved our GitHub repo and documentation so that our code is easier-to-use; (3) ensured that the analyses in the Supplementary Materials are cited in the main text.

Point-by-point responses:

Reviewer #1 (Remarks to the Author):

All my concerns have been well addressed.

We are glad that review 1 thinks all his/her comments have been well addressed.

Reviewer 2

First of all, I really appreciate that the authors put in the effort for all of the additional experiments. The authors have made many significant improvements to their manuscript. I have a few more straightforward but necessary concerns that need to be addressed before publication.

Major concerns

1) The authors have addressed my concerns.

2) I appreciate the improved documentation and github site. However, the train.py function is still not that simple to use. I would suggest making the script able to run from the command line such that `./train.py input_file1.txt input_file2.txt option1 option2` etc. Also, the readme should include an example of running the script on the example data. If the authors do not want to convert into an R or python package then there should be a clear command that can be run with clear documentation of the input files, required file formats, and various options that can be set when running train.py. All of the dependencies, including python packages, should be listed in the README file as well.

We agree that we should make the code easier-to-use. We have made a script(main.py) that is able to run from the command line and included clearer instructions in the README file. We also add a requirement.txt that lists all the python packages that need to be installed.

3) Please be sure to include this reasoning for omitting the tSNE tick labels and cite Tran et al. in that sentence. This explanation about UMAP is satisfactory.

We have added the reasoning for omitting the tSNE tick in the result section and cited Tran et al.

4) The authors have addressed my concerns.

5) The authors have addressed my concerns.

6) The authors have addressed my concerns.

7) I see in the supplementary information there are additional explanations of the experiments. However, these are never cited in the text. If the authors do not want to move more of this text into the main text, then it should be cited in the main text where appropriate. For example:

It was also important to evaluate the robustness of our algorithm to different initializations (see Supplementary Materials: Evaluation of robustness and reproducibility of SIDA).

We would like to thank for the comment. The information related to the additional explanations of the experiments (robustness and reproducibility experiment and the computational efficiency evaluation) are cited in the Discussion section.

Minor concerns

1) Please include this reasoning for using PCs instead of gene features somewhere in the main text methods Data preprocessing section. Please also comment on using PCs over genes in the discussion.

We appreciate this comment and agree with the reviewer that it is necessary to provide the reason of using PCs instead of genes features. We included the reasoning in the Methods section and commented on the this in the Discussion section.

2) Please include in the discussion some comments about how k was set to the number of cell types because this manuscript is primarily concerned with baseline comparisons.

Mention also that depending on the data there may be more subtypes of cells resulting in a greater number of clusters.

Thanks for this comment. We have included this in the Discussion section.

REVIEWERS' COMMENTS:

Reviewer #2 (Remarks to the Author):

The authors have addressed my concerns.

Response to Review

"Domain adaptation for supervised integration of scRNA-seq data"

Yutong Sun, Peng Qiu

We are glad that the second reviewer now thinks that we have addressed his/her comments. In this revision, we have made minor updates to satisfy the journal's formatting requirements. Thank you for your time in reviewing this work.